# Placental Remote Control of Fetal Metabolism: Trophoblast mTOR Signaling Regulates Liver IGFBP-1 Phosphorylation and IGF-1 Bioavailability

**DOI:** 10.3390/ijms24087273

**Published:** 2023-04-14

**Authors:** Fredrick J. Rosario, Anand Chopra, Kyle Biggar, Theresa L. Powell, Madhulika B. Gupta, Thomas Jansson

**Affiliations:** 1Division of Reproductive Sciences, Department of Obstetrics and Gynecology, University of Colorado Anschutz Medical Campus, Aurora, CO 80045, USA; 2Institute of Biochemistry, Carleton University, Ottawa, ON K1S 5B6, Canada; 3Section of Neonatology, Department of Pediatrics, University of Colorado Anschutz Medical Campus, Aurora, CO 80045, USA; 4Department of Biochemistry, University of Western Ontario, London, ON N6A 3K7, Canada; 5Department of Pediatrics, University of Western Ontario, London, ON N6A 3K7, Canada; 6Children’s Health Research Institute, University of Western Ontario, London, ON N6A 3K7, Canada

**Keywords:** casein kinase CK2, fetal growth retardation, pregnancy, human

## Abstract

The mechanisms mediating the restricted growth in intrauterine growth restriction (IUGR) remain to be fully established. Mechanistic target of rapamycin (mTOR) signaling functions as a placental nutrient sensor, indirectly influencing fetal growth by regulating placental function. Increased secretion and the phosphorylation of fetal liver IGFBP-1 are known to markedly decrease the bioavailability of IGF-1, a major fetal growth factor. We hypothesized that an inhibition of trophoblast mTOR increases liver IGFBP-1 secretion and phosphorylation. We collected conditioned media (CM) from cultured primary human trophoblast (PHT) cells with a silenced *RAPTOR* (specific inhibition of mTOR Complex 1), *RICTOR* (inhibition of mTOR Complex 2), or *DEPTOR* (activates both mTOR Complexes). Subsequently, HepG2 cells, a well-established model for human fetal hepatocytes, were cultured in CM from PHT cells, and IGFBP-1 secretion and phosphorylation were determined. CM from PHT cells with either mTORC1 or mTORC2 inhibition caused the marked hyperphosphorylation of IGFBP-1 in HepG2 cells as determined by 2D-immunoblotting while Parallel Reaction Monitoring-Mass Spectrometry (PRM-MS) identified increased dually phosphorylated Ser169 + Ser174. Furthermore, using the same samples, PRM-MS identified multiple CK2 peptides coimmunoprecipitated with IGFBP-1 and greater CK2 autophosphorylation, indicating the activation of CK2, a key enzyme mediating IGFBP-1 phosphorylation. Increased IGFBP-1 phosphorylation inhibited IGF-1 function, as determined by the reduced IGF-1R autophosphorylation. Conversely, CM from PHT cells with mTOR activation decreased IGFBP-1 phosphorylation. CM from non-trophoblast cells with mTORC1 or mTORC2 inhibition had no effect on HepG2 IGFBP-1 phosphorylation. Placental mTOR signaling may regulate fetal growth by the remote control of fetal liver IGFBP-1 phosphorylation.

## 1. Introduction

Intrauterine growth restriction (IUGR) is associated with functional changes in multiple fetal tissues, including skeletal muscle, the islet, and the liver [1,2,3]. These changes contribute to metabolic adaptations resulting in restricted fetal growth and predisposition for disease later in life. Although limited amino acid availability, hypoxia, oxidative stress, and epigenetic regulation have been implicated, the mechanisms underlying these functional changes in a range of fetal tissues in IUGR remain to be fully established.

Emerging evidence shows that factors secreted by the placenta are critical for normal fetal organ development. Using Stable Isotope Labelling by Amino Acids in Cell Culture, we have identified a wide array of secreted proteins in the conditioned media of a Primary Human Trophoblast (PHT) cell culture [4]. Moreover, serotonin synthesized by the placenta in mid-gestation is required for the normal development of the forebrain in mice, and maternal immune activation disrupts fetal neurodevelopment mediated by an increased placental serotonin secretion to the fetus [5,6]. In addition, the inhibition of placental O-GlcNAc transferase (OGT) constitutes the link between maternal stress and fetal hypothalamic gene expression changes and offspring neurocognitive development [7,8]. Importantly, using PHT cells, we recently reported that a mechanistic target of rapamycin (mTOR) is a negative regulator of serotonin synthesis and a positive regulator of OGT expression in the placenta [9], providing a possible link between placental mTOR signaling and fetal brain development.

mTOR regulates cellular metabolism, growth, and proliferation [10,11]. Placental mTOR is inhibited in IUGR in both women [12,13,14,15,16] and animal models [17,18,19,20,21]. Trophoblast mTOR signaling responds to diverse maternal nutritional and metabolic signals. For example, trophoblast mTOR is activated by insulin/IGF-1, glucose and amino acids [22], fatty acids [23], and folate [21,24,25] and inhibited by cortisol [26], adiponectin [20,27], malaria infection [15], and reduced uteroplacental blood flow [13]. Moreover, mTOR is a positive regulator of trophoblast amino acid [28,29,30,31] and folate transport [32] and mitochondrial respiration [33]. These data suggest that trophoblast mTOR signaling functions as a critical hub linking maternal nutrition, metabolism, and uteroplacental blood flow to placental function, fetal growth, and developmental programming.

Insulin-like growth factor 1 (IGF-1) is a key regulator of fetal growth [34,35,36], and the bioavailability of IGFs are regulated by six binding proteins (IGFBP-1–6) [37]. IGFBP-1, secreted by the fetal liver [38], is believed to be the predominant IGFBP during fetal life [39,40]. Mouse fetuses over-expressing IGFBP-1 are growth restricted [38,41,42], clearly demonstrating a cause-and-effect relationship between IGFBP-1 and fetal growth. The phosphorylation of IGFBP-1 leads to a 6–10-fold increase in its affinity for IGF-1 [43], resulting in an inhibition of IGF-1 function [43,44]. IUGR is associated with elevated fetal IGFBP-1 [45], and we have shown that IGFBP-1 phosphorylation at three specific residues is increased in human IUGR fetuses [46,47,48] and the decidua of IUGR pregnancies [49]. However, the mechanisms regulating fetal liver IGFBP-1 secretion and phosphorylation remain to be fully established. Here, we test the hypothesis that the inhibition of trophoblast mTOR increases fetal liver IGFBP-1 secretion and phosphorylation.

To study the effects of factors released by trophoblast cells that may be impacting fetal organs, such as fetal liver, we collected conditioned media (CM) from PHT cells in a culture (Figure 1). To experimentally modulate the activity of the mTOR pathway, we used siRNA to silence a regulatory-associated protein of mTOR: *RAPTOR* (resulting in specific inhibition of mTOR Complex 1, mTORC1); *RICTOR*, a key component of mTOR Complex 2 (mTORC2, resulting in mTORC2 inhibition); or DEP domain-containing mTOR-interacting protein *DEPTOR* (leading to the activation of both mTORC1 and mTORC2 Complexes). Subsequently, HepG2 cells (an established model for human fetal hepatocytes [50,51]) were cultured in CM from PHT cells, and IGFBP-1 secretion/site-specific phosphorylation and casein kinase CK2 activity were determined.

## 2. Results

### 2.1. CM from PHT Cells with mTORC1 Inhibition Increased IGFBP-1 Secretion in HepG2 Cells

Using immunoblotting, we showed that CM from PHT cells with mTORC1 (*RAPTOR* siRNA) moderately, but significantly, increased IGFBP-1 secretion in HepG2 cells (Figure 2A). In contrast, CM from PHT cells with mTORC2 inhibition (RICTOR siRNA) had no effect on IGFBP-1 secretion in HepG2 cells (Figure 2A).

### 2.2. Increased IGFBP-1 Phosphorylation in HepG2 Cells Incubated in CM from PHT Cells with Either mTORC1 or mTORC2 as Indicated by 2-D Immunoblot Analysis

We validated the effects of the CM from PHT cells with *RAPTOR* or *RICTOR* silencing on IGFBP-1 phosphorylation by comparing phosphoisoform profiles using 2-D immunoblotting. CM from HepG2 cells treated with CM from PHT cells silenced for *RAPTOR* (inhibition of mTORC1) or *RICTOR* (inhibition of mTORC2) both showed marked hyperphosphorylation of IGFBP-1 secreted from HepG2 cells compared to the control (non-coding siRNA). The 2-D immunoblots are displayed in Figure 2B (left panel), and the 3-D digital images (Figure 2C, right) are representations of the 2-D spot intensities. The *RAPTOR* siRNA (Figure 2B) exhibited six distinct spots (spot number 1 to 6), where spot 1 represents the least phosphorylated isoform, and spots 3–5 corresponded to variably increased phosphorylated isoforms, in which spot 6 indicated the most highly phosphorylated IGFBP-1 isoform. The corresponding 3-D digital image (Figure 2C) showed that the highly phosphorylated isoforms were the most abundant, whereas the less phosphorylated isoforms had a diminished abundance. Similarly (Figure 2B,C), *RICTOR* showed significantly increased phosphorylation (spots 2–5), showing that the incubation of HepG2 cells in CM from PHT cells with mTORC1 inhibition and incubation with CM from PHT cells mTORC2 inhibition resulted in similar patterns of IGFBP-1 hyperphsophorylation. We propose that the marked shift in the IGFBP-1 phosphoisoform profile to the left is caused by an increased net protein charge (pI) due to an increased phosphorylation level.

### 2.3. Incubation of HepG2 Cells in CM from PHT Cells with mTORC1 Increased Site-Specific IGFBP-1 Phosphorylation

To determine the site-specific IGFBP-1 phosphorylation, we performed immunoblotting with antibodies previously validated to recognize pSer101, pSer119, and pSer169 [48,52]. As shown in Figure 3A–C, CM from PHT cells with mTORC1 inhibition, but not CM from PHT cells with mTORC2 inhibition, markedly increased Ser101, Ser119, and Ser169 phosphorylation of IGFBP-1 secreted from HepG2 cells (Figure 3A–C).

### 2.4. Parallel Reaction Monitoring Mass Spectrometry (PRM-MS) Analysis Identified Increased IGFBP-1 Phosphorylation at pSer169/pSer174 Following Incubation in CM from PHT Cells with mTOR Inhibition

To determine relative levels of dual IGFBP-1 phosphorylation at pSer169 and pSer174, we performed PRM-MS, as no antibodies were available to assess this modification. Following co-immunoprecipitation, PRM-MS readily detected unmodified and dual pSer169/pSer174 IGFBP-1 peptides (Figure 4A–C). In addition, both CM from PHT cells with mTORC1 or mTORC2 inhibition resulted in a several-fold increase in IGFBP-1 pSer169/pSer174 levels relative to the control (Figure 4D).

### 2.5. HepG2 Cells Incubated in CM from PHT Cells with mTORC1 or mTORC2 Inhibition Activated CK2 Activity

We then assessed casein kinase CK2, which has been shown to be the major kinase phosphorylating IGFBP-1 in HepG2 cells [53,54]. From PRM-MS, CK2 peptides co-immunoprecipitated with IGFBP-1 were readily detected in all samples (Figure 5A). In addition, we found that CM from PHT cells with mTORC2 inhibition increased the relative levels of co-immunoprecipitated CK2 (Figure 5B). As shown in Figure 5C, CM from PHT cells with either mTORC1 or mTORC2 inhibition activated CK2 activity in HepG2. This finding is in line with the increased IGFBP-1 phosphorylation under these conditions, as indicated by the immunoblot and PRM-MS analysis (Figure 2, Figure 3 and Figure 4).

### 2.6. IGF-1R Autophosphorylation Decreased in HepG2 Cells Incubated in CM from PHT Cells with mTORC1 or mTORC2 Inhibition

The functional effects of phosphorylated IGFBP-1 secreted from HepG2 cells incubated in CM collected from PHT cells treated with the control (non-coding siRNA), *RAPTOR*, or *RICTOR* siRNA was determined (Figure 6A) by assessing the relative inhibitory effects on IGF-1 induced IGF-1 receptor autophosphorylation in P6 cells (*n* = 3) (methodology described in Appendix A). Our data in Figure 6A (Lane 2) shows a marked increase in IGF-1Rβ autophosphorylation in P6 cells treated with IGF-1 alone, serving as a positive control.

Notably, the control CM constitutes both phosphorylated and non-phosphorylated IGFBP-1 forms. Accordingly, when P6 cells were treated with IGF-1 in complex with CM from the control (Figure 6A, Lane 3), IGF-1R autophosphorylation was markedly inhibited compared to IGF-1 alone (positive control) (Figure 6A, Lane 2). Importantly, in P6 cells treated with IGF-1 in complex with CM from PHT cells with either *RAPTOR* (Figure 6A, Lane 4) or *RICTOR (*Figure 6A, Lane 5) siRNA, IGF-1Rβ autophosphorylation was further reduced. Thus, we demonstrate that the increased phosphorylation of IGFBP-1 in HepG2 cells following incubation with CM from PHT cells with either mTORC1 or mTORC2 inhibition (Figure 2, Figure 3 and Figure 4) functionally is associated with decreased IGF-1-mediated IGF-1R activation. We propose that these changes are a result of an enhanced affinity of phosphorylated IGFBP-1 for IGF-1 following incubation in CM from PHT cells with mTOR inhibition.

The inhibition of IGF-1R in P6 cells was also tested using trophoblast cell media from the control, *RAPTOR*, and *RICTOR* knockdown in the absence of externally added IGF-1 (Figure 6B) and showed no inhibitory effects on IGF-1R autophosphorylation, serving as a negative control.

### 2.7. DEPTOR Silencing Markedly Inhibited IGFBP-1 Phosphorylation

DEPTOR is an endogenous inhibitor of both mTORC1 and 2, and silencing DEPTOR activates mTOR signaling [55]. Although the secretion of IGFBP-1 was not affected, phosphorylation was markedly inhibited at all three (Ser101, 119, and 169) phosphorylation sites in HepG2 cells treated with CM from PHT cells with DEPTOR silencing while relatively smaller decrease in IGFBP-1 secretion (Figure 7A–D).

### 2.8. No Change in IGF-1 Secretion from PHT Cells with mTORC1 or mTORC2 Inhibition

The effect of CM from PHT cells on IGFBP-1 secretion and phosphorylation in HepG2 cells could be confounded if *RAPTOR* or *RICTOR* silencing regulates the secretion of IGF-1 in PHT cells. IGF-1 secretion was determined in CM from PHT cells with the control, *RAPTOR*, and *RICTOR* siRNA (Figure 8A) and in CM from HepG2 cells after treatment (Figure 8B) with CM from PHT cells with the control, *RAPTOR*, and *RICTOR* siRNA. However, neither *RAPTOR* nor RICTOR silencing impacted PHT or HepG2 cell IGF-1 secretion (Figure 8A,B).

### 2.9. CM from RAPTOR- or RICTOR-Silenced PHT Cells Does Not Affect mTORC1 or mTORC2 Activity in HepG2 Cells

We have previously demonstrated that mTOR signaling is a key regulator of IGFBP-1 phosphorylation in HepG2 cells [48,55] and primary fetal baboon hepatocytes [48]. Thus, we explored the possibility that HepG2 mTOR signaling may mediate the effect of CM from *RAPTOR*- or *RICTOR*-silenced PHT on IGFBP-1 phosphorylation. However, we found that CM from *RAPTOR*- or *RICTOR*-silenced PHT cells does not affect mTORC1 or C2 activity in HepG2 cells (Figure 8C,D) as reflected by no changes in phosphorylation of its functional readouts p4E-BP1 ^Thr70^ (mTORC1) (Figure 8C) and pAKT ^Ser473^(mTORC2). (Figure 8D).

### 2.10. CM from Non-Trophoblast Cells with mTORC1 or mTORC2 Inhibition Did Not Affect HepG2 IGFBP-1 Phosphorylation

To explore to what extent the marked effects of PHT cell CM on HepG2 cell IGFBP-1 phosphorylation is specific to trophoblast cells, we examined the impact of CM collected from renal carcinoma cells in which *RAPTOR* or *RICTOR* was silenced. As shown in Figure 9, CM from non-trophoblast cells with mTORC1 or mTORC2 inhibition did not affect HepG2 IGFBP-1 secretion or phosphorylation.

## 3. Discussion

The mechanisms mediating the restricted growth in IUGR are poorly understood. mTOR signaling functions as a placental nutrient sensor, indirectly influencing fetal growth by regulating placental functions like protein synthesis, nutrient transport, and oxidative phosphorylation. In this study, we provide evidence to support the model (Figure 10) that placental mTOR signaling directly influences the endocrine and metabolic function of fetal tissues, specifically by modulating the bioavailability of IGF-1, mediated by signals released into fetal circulation.

IGFBP-1 is the predominant IGF-1 binding protein in fetal life, and phosphorylation of the protein increases its affinity for IGF-1 6–10-fold [43] and makes IGFBP-1 more resistant to proteolysis [44,56]. Functionally, phosphorylation increases the capacity of IGFBP-1 to inhibit IGF-1-stimulated cell proliferation, DNA synthesis, amino acid transport, and apoptosis [53,57,58]. Furthermore, we reported that hepatic IGFBP-1 phosphorylation that was induced in response to hypoxia caused a profound increase in its affinity for IGF-1, resulting in a marked inhibition of IGF-1-dependent cellular proliferation [59,60]. Thus, the regulation of fetal liver IGFBP-1 phosphorylation represents a powerful mechanism to alter the trajectory of fetal growth mediated by changes in IGF-1 bioavailability.

The 2-D immunoblotting based on isoelectric points of IGFBP-1 demonstrates that CM from both *RAPTOR* and *RICTOR* siRNA-treated PHT cells caused marked shifts in spots toward the acidic pH. These observations indicate that the phosphorylated amino acid residues contribute to the IGFBP-1 net protein charge showing pronounced increases in IGFBP-1 phosphorylation.

Using targeted multiple reaction monitoring-mass spectrometry (MRM-MS) for relative quantification of IGFBP-1 phosphorylation, we have previously shown Ser174 phosphorylation singly and/or combined with Ser169 is an mTOR-sensitive IGFBP-1 phosphorylation potentially involved in IGF-1 binding [55]. Further, using PRM-MS analysis, we demonstrated increased dual (Ser174/Ser169) IGFBP-1 phosphorylation in the plasma of mothers carrying an IUGR fetus [49].

In this study using PRM-MS, we demonstrate that the increased phosphorylation of IGFBP-1 secreted by HepG2 cells incubated in CM from PHT cells with *RICTOR* silencing involves Ser174 phosphorylation combined with Ser169. Based on molecular modeling and mapping the proximity for pSer174 binding with IGF-1, we earlier proposed that phosphorylation at residue Ser174 likely affects the affinity of the IGFBP-1 to bind IGF-1 [55]. Furthermore, we showed that CM from PHT cells with the inhibition of mTORC1- or mTORC2-activated HepG2 CK2, an enzyme playing a key role in phosphorylating IGFBP-1, resulted in IGFBP-1 hyperphosphorylation. In this study, IGFBP-1 hyperphosphorylation also decreased IGF-1 bioavailability, which had functional effects on IGF-1 action. Although we have provided evidence previously that CK2 is a major contributor to IGFBP-1 phosphorylation, including in response to changes in mTOR signaling, we often find that measured changes in CK2 activity are lower than the measured changes in IGFBP-1 phosphorylation. There may be several reasons for this discrepancy. First, it is possible that the relationship between CK2 activity and IGFBP-1 phosphorylation is not 1:1 and that small changes in CK2 activity are related to larger changes in IGFBP-1 phosphorylation. Second, there may be additional kinases, such as PKA or PKC, contributing to IGFBP-1 phosphorylation; third, it cannot be excluded that our approach to measure CK2 activity underestimates the actual activity.

These changes could not be replicated by incubating HepG2 cells in CM from renal carcinoma cells where *RAPTOR* or *RICTOR* had been silenced, suggesting these signals are specific to trophoblast cells. Because mTOR regulates IGFBP-1 phosphorylation in HepG2 cells [48,55] and primary fetal baboon hepatocytes [48], we tested the hypothesis that signals secreted from cultured trophoblast cells with mTORC1 or mTORC2 inhibition impacted HepG2 mTOR signaling. However, CM from *RAPTOR*- or *RICTOR*-silenced PHT cells did not affect mTORC1 or C2 activity in HepG2 cells. Similarly, we found no evidence that changes in IGF-1 concentrations in the PHT CM influenced our results.

Although the identity of the factor(s) linking trophoblast mTOR signaling to regulation of IGFBP-1 in the fetal liver remains to be established, the signal may be a protein secreted from the trophoblast. mTOR signaling is a master regulator of protein synthesis, and trophoblast mTOR may regulate the secretion of proteins from the placenta into the maternal and/or fetal circulation. Using Stable Isotope Labeling with Amino Acids in Cell Culture, we identified a total of 1344 proteins secreted by cultured PHT cells, most of which have not previously been reported to be secreted by the human placenta or trophoblast [4]. Importantly, using the SOMALOGIC proteomic platform, we recently reported that the human term placenta secretes 341 proteins into fetal circulation [61].

Another possibility is that exosomes mediate the remote control of trophoblast mTOR signaling on liver IGFBP-1 secretion and phosphorylation. Exosomes are small (~40–100 nm) nanovesicles produced via the endosomal pathway and released upon the fusion of multivesicular bodies with the plasma membrane [62]. Exosomes transport bioactive proteins, mRNAs, and miRNAs and play a central role in cell-to-cell communication in normal physiology and disease [63,64,65], including in pregnancy [66,67]. Importantly, mTORC1 regulates exosome release in mouse embryo fibroblasts, cell lines, and in vivo in mice [68], and mesenchymal stem cell-derived exosomes enhance β-cell mass and insulin production in the pancreas of diabetic animals mediated by PDX1 [69]. Furthermore, the human placenta secretes exosomes into the maternal [66,67,70,71,72,73] and fetal circulation [74]. Mouse studies have shown that miRNA is trafficked from the placenta into both the maternal and fetal circulations [75], consistent with the possibility that the placenta regulates the development and function of fetal organs mediated by exosomal miRNA.

Emerging evidence suggests that trophoblast mTOR signaling represents a critical hub in the overall homeostatic control of fetal growth, adjusting the fetal growth trajectory according to changes in maternal nutrition and metabolism and uteroplacental blood flow [76,77]. In support of this model, placental mTOR signaling is inhibited in IUGR and activated in fetal overgrowth. Specifically, placental mTOR signaling is inhibited in IUGR in women [12,13,14,15,28] and animal models of IUGR [17,18,20,21]. Importantly, placental mTOR signaling has been shown to be inhibited prior to its reduction in fetal growth [17], consistent with the possibility that changes in placental mTOR signaling are a cause rather than a secondary consequence of abnormal fetal growth. mTOR is a master regulator of placental function. For example, trophoblast mTOR inhibits autophagy [78]. Moreover, mTOR signaling is a positive regulator of trophoblast amino acid [28,29,30,31] and folate transport [32] and mitochondrial respiration [33]. It is therefore believed that trophoblast mTOR signaling influences fetal growth indirectly by regulating placental function [76,77].

This report provides evidence that mTOR signaling in the trophoblast directly regulates fetal growth by modulating the bioavailability of IGF-1, a key fetal growth factor [34,35], providing novel mechanistic information on the role of the placenta in regulating fetal metabolism and growth. Because the placenta—in contrast to the fetus—is easily accessible from the maternal circulation and an array of approaches to specifically target the placenta with, for example, nanoparticles is currently under development, we speculate that the gene targeting of placental mTOR may provide an avenue to correct abnormal fetal growth in the future. The main limitation of the study is that the data is based on in vitro studies, and our findings need to be confirmed in vivo using a placental-specific targeting of mTOR.

## 4. Materials and Methods

### 4.1. Human Placenta Collection

We recruited healthy term (>37 weeks of gestation) pregnant women delivered by Caesarean following written informed consent. Exclusion criteria included concurrent maternal disease; pregnancy complications like gestational diabetes, pregnancy-induced hypertension, and preeclampsia; smoking; and the use of illicit drugs. The Institutional Review Boards at the University of Texas Health Science Center at San Antonio (HSC20100262H) and the University of Colorado Hospital (number 14-1073) approved this protocol.

### 4.2. Isolation and Culture of Primary Human Trophoblast PHT Cells

A well-established method involving sequential trypsin digestion and Percoll centrifugation [30,79] was used to isolate and culture primary human trophoblast (PHT) cells. Briefly, isolated PHT cells were plated at 1.5 × 10^6^ per well in six-well plates and cultured in a 1:1 mixture of Dulbecco’s modified Eagle’s medium (DMEM, 25 mM glucose) and Ham’s F12 medium (10 mM glucose) with 2 mM glutamine, 50 μg/mL gentamicin, 60 μg/mL penicillin, 100 μg/mL streptomycin, and 10% fetal bovine serum. Media was changed at 18 h, followed by daily media changes.

### 4.3. RNA Interference-Mediated Silencing (siRNA) in PHT Cells

PHT cells were plated at 2.5 × 10^6^ per well in six-well plates and cultured as described above. At 18 h, culture, small interference RNAs (siRNAs) (Sigma) targeting *RAPTOR* (100 nm; sense, 5′CAGUUCACCGCCAUCUACA, or SASI_Hs01_00048382); *RICTOR* (100 nm; sense, 5′ CGAUCAUGGGCAGGUAUUA, or SASI_Hs01_00223573); *DEPTOR* (SASI_1297010-H/5582, 1297011-H); or a non-coding control (100 nM; sense: 5′GAUCAUACGUGCGAUCAGATT) were added to the media along with DharmaFECT transfection reagent (Thermo Fisher, Rockford, IL, USA) and incubated for 24 h [80]. Cells were then washed and subsequently cultured in standard media. CM was collected from PHT cells in culture between 66 and 90 h and stored at −80 °C.

### 4.4. HepG2 Cell Culture

Human hepatocellular carcinoma cells (HepG2) (American Type Culture Collection (ATCC), Manassas VA) were cultured in Dulbecco’s modified Eagle’s medium with nutrient mixture F-12 (DMEM/F-12) supplemented with 10% fetal bovine serum (FBS) (Invitrogen Corp., Carlsbad, CA) at 37 °C in 20% O_2_ and 5% CO_2_ [48,59]. Following treatments, cell media and cell lysate were prepared as described [48] and stored at −80 °C.

### 4.5. Incubation of Cultured HepG2 Cells in CM from PHT Cells

HepG2 cells were plated and grown to ~70% confluency (overnight). Cells were rinsed with fresh DMEM/F-12 (FBS-free). CM (800 uL) collected from PHT cells silenced with *RAPTOR*, *RICTOR*, *DEPTOR*, or non-coding siRNA were topped up to 1 mL total volume by the addition of 200 uL of fresh FBS free DMEM/F-12 media. HepG2 cells were subsequently incubated for 24 h in CM collected from PHT cells treated with either control (non-coding), *RAPTOR* (inhibition of mTORC1), *RICTOR* (inhibition of mTORC2), or *DEPTOR* (activation of mTORC1 and C2) siRNA. Cell media and cell lysate from HepG2 cells incubated with CM from respective PHT treatments were collected after 24 h and stored at −80 °C until further analysis.

### 4.6. SDS-PAGE and Western Blotting

Equal amounts of cell lysate protein (35–50 μg) from HepG2 cells were separated by SDS-PAGE to determine total expression and phosphorylation of p70-S6K (Thr389), Akt (Thr308 and Ser473), and IGF-1Rβ (Tyr1135), as well as total expression levels of siRNA target proteins and β-actin. In addition, IGFBP-1 secretion and phosphorylation (Ser101, Ser119, and Ser169) by HepG2 cells were determined using an equal volume of cell media (30–40 μL). Nitrocellulose membranes from immunoblot analyses were blocked with 5% skim milk or 5% BSA in Tris-buffered saline (TBS) plus 0.1% Tween-20 for 1 h. All primary antibodies were obtained from Cell Signaling Technologies (Beverly, MA, USA) except for monoclonal anti-human IGFBP-1 (mAb 6303) (Medix Biochemica, Kauniainen, Finland) and pre-validated custom phospho-site-specific IGFBP-1 polyclonal antibodies targeting pSer101, pSer119, and pSer169 (generated at YenZyme Antibodies LLC, San Francisco, CA, USA). Primary antibodies were all used at a dilution of 1:1000, and peroxidase-labeled goat-anti mouse or goat-anti rabbit antibodies (1:10,000, BioRad Laboratories Inc., Mississauga, ON, Canada) were used as secondary antibodies. Band intensities were determined using densitometry in Image Lab (Beta 3) software (BioRad). In the absence of a recognized loading control for secreted proteins (equivalent to actin for non secreted protein), we used an equal volume of cell media [55,59] as a loading control.

### 4.7. Two-Dimensional (2-D) Immunoblot of IGFBP-1

HepG2 CM from either control (non-coding siRNA), *RAPTOR*-silenced (inhibition of mTORC1), *RICTOR*-silenced (inhibition of mTORC2), or *DEPTOR*-silenced (Activation of mTORC1 and C2) PHT cells were analyzed on 2-D Western immunoblotting to separate the IGFBP-1 phospho-isoform based on isoelectric point (pI). The 2-D immunoblot analysis with HepG2 cell media (~150 μL) was performed using polyclonal IGFBP-1 antibody (a kind gift from Dr. Rob Baxter, Northern Clinical School, Kolling Institute, St Leonards, Australia). Subsequent three-dimensional (3-D) image analysis of the 2-D spot intensity and the phosphoisoform profiles were created as described previously [47,59,80] using Progenesis 200 (PG200; Non Linear Dynamics, Durham, NC, USA).

### 4.8. Parallel Reaction Monitoring-Mass Spectrometry (PRM-MS)

#### 4.8.1. Immunoprecipitation of IGFBP-1

Equal aliquots of CM samples from HepG2 cells treated with CM from *RAPTOR*- or *RICTOR*-silenced PHT cells or control were used for PRM-MS analysis. Prior to immunoprecipitation, total IGFBP-1 was validated with Western blot (not shown). PRM-MS of IGFBP-1 phosphopeptides was run essentially in the same manner as previously described [81], whereby three biological replicates of respective CM were pooled and used for immunoprecipitation. In brief, specific samples were immunoprecipitated (IP) with IGFBP-1 mouse monoclonal anti-human (mAb) 6303 antibody (Medix Biochemica, Kauniainen, Finland) using Protein A Sepharose beads (50 µL, 50% slurry; GE Healthcare Bio-Sciences AB, Uppsala, Sweden) as described previously [81].

#### 4.8.2. Parallel Reaction Monitoring Mass Spectrometry (PRM-MS) Analyses of IGFBP-1 Peptides and CK2 co-Immunoprecipitation

The IP samples were digested as described below for PRM-MS analysis. The quantification of site-specific phosphorylation of both IGFBP-1 and CK2 was performed using PRM-MS approaches designed to monitor the presence of peptide-specific mass [m/z] and peptides post-translationally modified at specific sites by phosphorylation. In addition, CK2 peptides co-immunoprecipitation (co-IP) with IGFBP-1 were identified with PRM-MS and tested for their activation status via autophosphorylation.

Beads from the IP containing bound IGFBP-1 proteins were first exchanged into trypsin digestion buffer (50 mM (NH_4_) HCO_3_, pH 7.8) and then digested with 10 ng of sequencing grade trypsin (Promega, V511A) overnight at 37 °C. Half of each sample was then diluted with 2X Asp-N reaction buffer (NEB, B8104S) and digested with 10 ng of Asp-N (NEB, P8104S) overnight at 37 °C. The resultant peptides were desalted with C18-ZipTip (Millipore Sigma, ZTC18S096), dried by SpeedVac, and resuspended in 20 μL MS-grade water containing 0.1% formic acid. PRM-MS was performed at the John L. Holmes Mass Spectrometry Facility at the University of Ottawa. Briefly, peptides were chromatographically separated on a Thermo Scientific™ Acclaim PepMap™ 100 precolumn (2 cm × 100 μm ID, C18, 5 μm, 100 Å) and a Thermo Scientific™ Acclaim™ PepMap™ RSLC C18 column (15 cm × 75 μm ID, 3 μm, 100 Å) using an EASY nLC II System (Thermo Fisher Scientific). First, the peptides were loaded onto the columns for 100 min at a flow rate of 0.25 μL/min. Separation occurred using multiple linear gradients of water + 0.1% formic acid (A) and acetonitrile + 0.1% formic acid (B) solutions (0 to 6% B for 2 min, 6 to 43% B for 70 min, 43 to 100% B, for 4 min, washing for 10 min at 100% B, then 100 to 10% B for 4 min, and washing for 10 min at 100% A). Next, eluted peptides were directly electrosprayed (Thermo Scientific™ EASY spray, Waltham, Massachusetts, U.S.) into a Q-Exactive Plus hybrid quadrupole-orbitrap mass spectrometer, where PRM scanning was performed and guided by an isolation list (i.e., peptide sequences and m/z ratios for IGFBP-1 and CK2 peptides) developed in Skyline software [82].

Chromatographs depict transitions ions detected for a given parent ion (i.e., peptide), where the retention time specifies the elution of the parent ion from the column just prior to MS. To quantify IGFBP-1 dual pS169/pS174 phosphorylation, the total peak area was derived using Skyline software and normalized to that of an unmodified IGFBP-1 internal peptide (NH_2_-DNFHLMAPSEE-COOH) to account for any slight differences in protein abundance. In the same manner, CK2 pY182 phosphorylation levels were quantified using peak areas of modified phosphopeptide (NH_2_-DWGLAEFY[Pho]HPGQEYNVR-COOH) and unmodified CK2 internal peptide (NH_2_-DVNTHRPREYWDYESHVVEWGNQ-COOH). To quantify levels of CK2 and PKC co-immunoprecipitated with IGFBP-1, the total peak area for multiple peptides along the CK2 and PKC protein sequences were summed for each sample and normalized to that of the control.

### 4.9. IGF-1 Receptor Activation Assay

To examine whether the alterations in IGFBP-1 phosphorylation functionally alter IGF-1 receptor (IGF-1R) autophosphorylation, P6 cells (a BALB/c3T3 cells derivative, gift from Dr. R. Baserga, Thomas Jefferson University) were used. P6 cells are immortalized mouse embryonic fibroblast cells that overexpress human IGF-1R but do not express IGF-1 [83]. We used a well-established protocol developed in our lab [48,52]. In brief, P6 cells were cultured in DMEM/F12 with sodium pyruvate supplemented with 10% FBS. The P6 cell bioassay was performed in FBS-free conditions. Aliquots of HepG2 cell media from various treatments conducted using control (non-coding siRNA), *RAPTOR*-silenced (inhibition of mTORC1), or *RICTOR*-silenced (inhibition of mTORC2) CM from PHT cells containing equal concentrations of IGFBP-1 (250 ng IGFBP-1) were buffer-exchanged to P6 cell media using Amicon Ultra-0.5 mL Centrifugal Filter Units (Millipore, Darmstadt, Germany) per manufacturer’s instructions. As illustrated in Appendix A, CM from HepG2 cells from respective PHT cell treatments were subsequently incubated with rhIGF-1 (25 ng/mL) for two hours at room temperature. IGF-1 (10 ng) served as a positive control. P6 cells were then treated for 10 min with the P6 media containing IGFBP-1: IGF-I complexes or with IGF-1 alone (positive control) and were used to treat serum-starved P6 cells (75% confluent) in culture dishes for 10 min. The P6 cells were collected in a lysis buffer (Cell Signaling Technology, Danvers, Massachusetts) containing protease inhibitors and phosphatase inhibitor cocktails (Sigma Aldrich). Subsequently, lysed samples were separated using SDS-PAGE, and immunoblot analysis was performed to assess IGF-1R autophosphorylation using phospho-site specific IGF-1Rβ (Tyr1135) primary antibody. The membrane was stripped (Pierce Chemical Co) and re-probed with an antibody for total anti-IGF-1R (Santa Cruz Biotechnology, Inc., Santa Cruz, CA, USA). The densitometry of the phosphorylated IGF-1R band was determined by Image Lab (Beta 3) software (Bio-Rad Laboratories) and was normalized to the band intensity of total IGF-1Rβ. The data used for final analysis were collected in triplicate from three biological experiments.

### 4.10. Casein Kinase CK2 Activity Assay

CM collected from HepG2 cells treated with CM from control (non-coding siRNA), *RAPTOR*-silenced (inhibition of mTORC1), or *RICTOR*-silenced (inhibition of mTORC2) PHT cells were analyzed for CK2 activity using P^32^ as described previously [47,59,80]. In addition, the CK2 substrate peptide RRRDDDSDDD (DSD) (100 μM) was used to measure CK2 activity [84].

### 4.11. Enzyme-Linked Immunosorbent Assay (ELISA) for IGF-1 in CM Samples of HepG2 Cells

HepG2 cells treated for 24 h with CM collected from PHT cells either control (non-coding siRNA), *RAPTOR*-silenced (inhibition of mTORC1), and *RICTOR*-silenced (inhibition of mTOR C2) PHT cells were used for the quantitative measurement of IGF-1 using human IGF-1 ELISA kit (Abcam). IGF-1 was also measured in CM from PHT cells with either control, *RAPTOR*-, or *RICTOR*-silenced PHT cells before treatment of HepG2 cells.

### 4.12. Non-Trophoblast Renal Carcinoma Cells

Renal carcinoma A498 cells were obtained from the ATCC and maintained in DMEM supplemented with 10% heat-inactivated fetal bovine serum at 37 °C in a humidified 5% CO2 atmosphere. Cells were transfected with pooled siRNA reagent (Thermo Fisher) with the Amaxa Nucleofector system for *RAPTOR* and *RICTOR* knockdown according to the manufacturer’s protocol. Samples of CM were harvested at 72 h following transfection. A non-targeting scramble siRNA pool was used as negative control (Thermo Fisher) [85].

### 4.13. Data Presentation and Statistics

Data were analyzed using GraphPad Prism 5 (Graph Pad Software Inc., CA). Three replicates were analyzed for each treatment condition, including the control treatment. For each quantified protein, the mean density of the control sample bands was assigned an arbitrary value of 1, and averaged densitometry values for each treatment were expressed relative to this mean. We employed a one-way analysis of variance with Dunnett’s Multiple Comparison Post-Test and expressed results as the mean ± SEM. Significance was accepted at * *p* < 0.05.

## Figures and Tables

**Figure 1 ijms-24-07273-f001:**
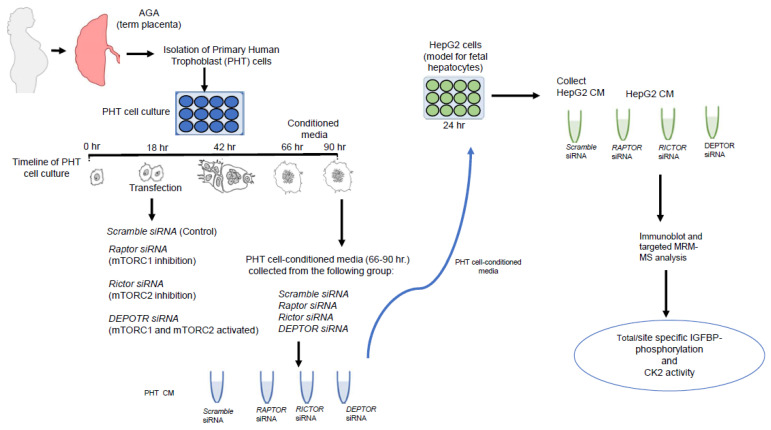
Schematic representation of the experimental design.

**Figure 2 ijms-24-07273-f002:**
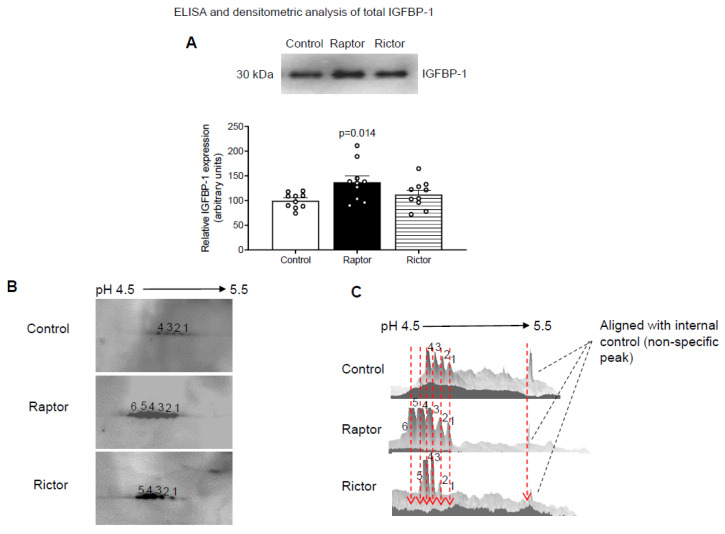
Analysis of IGFBP-1 secretion and IGFBP-1 phosphorylation by 1-D and 2-D immunoblot analysis using CM from HepG2 cells incubated with CM from PHT cells with the control, *RAPTOR*, and *RICTOR* siRNA. For a densitometric analysis (**A**), proteins in equal aliquots of CM from HepG2 cells (7 µL) treated with CM from PHT cells with the control (non-coding siRNA), *RAPTOR,* and *RICTOR* siRNA were separated and probed using anti-human IGFBP-1 monoclonal antibody (mAb 6303). The data were normalized to the control (average arbitrarily set to 100) with Mean + SEM, analyzed with one-way ANOVA, and corrected with Bonferroni multiple comparison test; *p* < 0.05 was considered significant. (**B**) Equal aliquots of HepG2 CM (100 µL) from PHT cells with Control (non-coding siRNA), *RAPTOR,* and *RICTOR* siRNA were exchanged against a rehydration buffer and loaded onto a 7 cm IPG strip followed by 2-D gel electrophoresis and a Western blot using anti-human IGFBP-1 polyclonal antibody. The spots on the blots (**B**) corresponded to IGFBP-1 phosphoisoform in CM from HepG2 cells treated with CM from PHT cells with the control (non-coding siRNA), *RAPTOR*, and *RICTOR* siRNA. The corresponding 3-D view (**C**) of IGFBP-1 phosphoisoform in the control, *RAPTOR*, and *RICTOR* is represented in the adjacent figure on the right.

**Figure 3 ijms-24-07273-f003:**
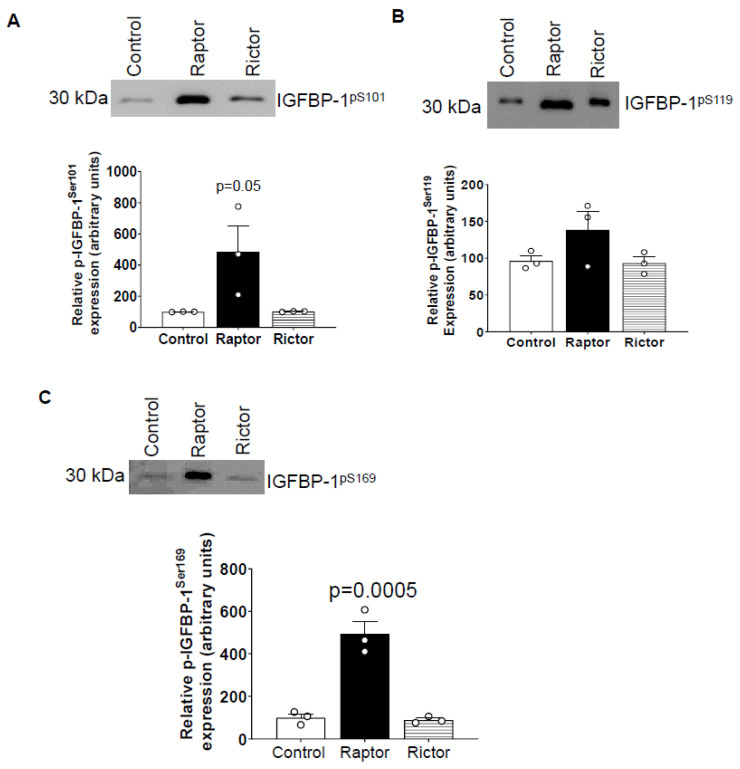
Site-specific IGFBP-1 phosphorylation in CM from HepG2 cells treated with CM from PHT cells with the control (non-coding siRNA), *RAPTOR,* and *RICTOR* siRNA. Equal aliquots of CM (μL) from HepG2 treated with CM from PHT cells with the control (non-coding siRNA), *RAPTOR*, and *RICTOR* siRNA was used to determine site-specific IGFBP-1 phosphorylation. CM (40 µL) was loaded on 1-D gels for Western blot for pSer101 (**A**), pS119 (**B**), and pS169 (**C**) IGFBP-1. The band intensity of each sample was normalized to the control (average arbitrarily set to 100). Data were presented in the bar figure as Mean + SEM, analyzed using one-way ANOVA, and corrected with Bonferroni Multiple Comparison tests; *p* < 0.05 was considered significant.

**Figure 4 ijms-24-07273-f004:**
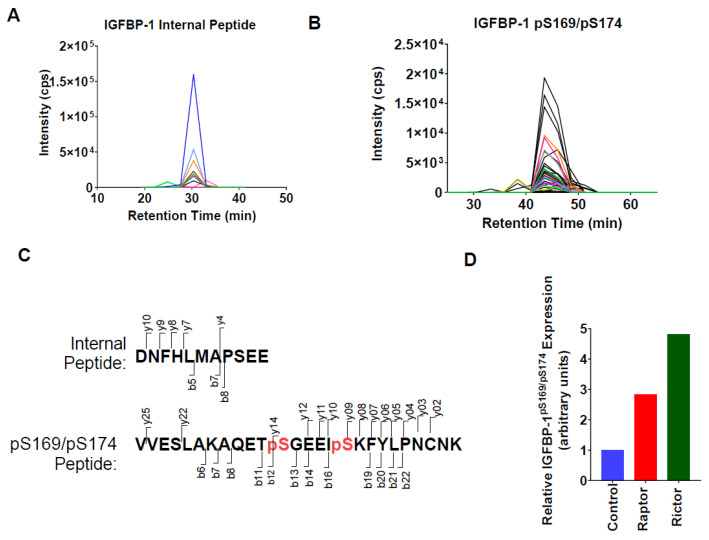
Parallel Reaction Monitoring Mass Spectrometry (PRM-MS) analysis identified increased IGFBP-1 phosphorylation at pSer169/pSer174 following incubation in CM from PHT cells with mTOR inhibition. To determine relative levels of dual IGFBP-1 phosphorylation at pSer169 and pSer174, PRM-MS was performed, as no antibodies were available to assess this modification. Detection of IGFBP-1 (**A**) internal (*m*/*z* = 645.2770, z = +2) and (**B**) pS169/pS174 (*m*/*z* = 1015.4623, z = +3) peptides by PRM-MS. Displaying chromatograms of individual fragment ions. (**C**) The red font indicates the peptide sequences and fragmentation patterns corresponding to panels A and B. Serine phosphorylation sites. (**D**) The peak area of pS169/pS174 peptides normalized to the peak area of the internal peptide control, shown as relative to the control (non-coding) siRNA.

**Figure 5 ijms-24-07273-f005:**
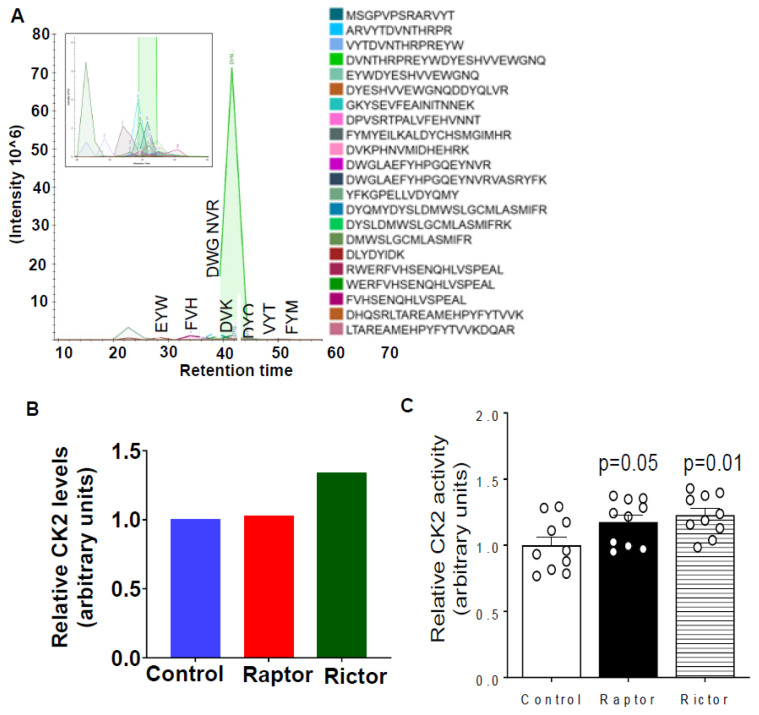
HepG2 cells incubated in CM from PHT cells with mTORC1 or mTORC2 inhibition increased CK2 activity. (**A**) Detection of CK2 peptides with PRM-MS; displaying chromatograms of individual peptides. The inset displays lower-intensity peptides. (**B**) Relative CK2 levels co-immunoprecipitated with IGFBP-1 from HepG2 CM after treatment with CM from PHT cells with either the control, *RAPTOR* (Rap), or *RICTOR* (Ric) siRNA. The total peak area for each treatment was normalized to that of the control. (**C**) CM from PHT cells with either mTORC1 or mTORC2 inhibition activated CK2 activity in HepG2.

**Figure 6 ijms-24-07273-f006:**
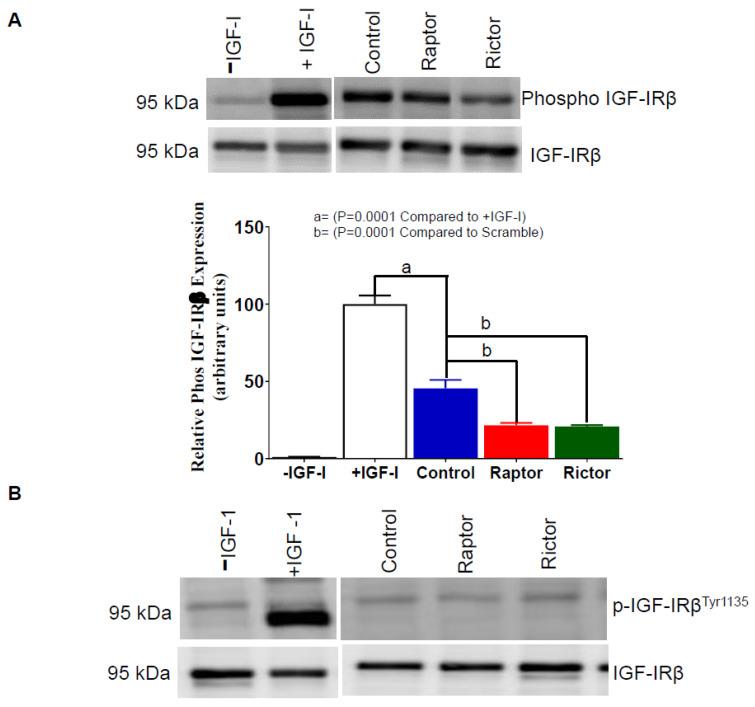
Inhibition of IGF-1R autophosphorylation in P6 cells using CM from HepG2 Cells treated with CM from PHT cells with *RAPTOR* and *RICTOR* siRNA. The ability of IGFBP-1 secreted from HepG2 cells treated with CM from PHT cells with the control, *RAPTOR,* and *RICTOR* siRNA to inhibit IGF-1 receptor autophosphorylation in P6 cells. (**A**) An equal amount of P6 cell lysate protein (25 µg) was loaded on 1-D gels for a Western blot using an antibody against p-IGF-1Rβ ^Tyr1135^. (**B**) Aliquots (1 mL) of HepG2 CM was used to treat a confluent culture of P6 cells in the absence of IGF-1 as a negative control while a serum-free sample containing 10 ng IGF-1 was a positive control. An equal protein amount (25 µg) of treated P6 cell lysate was loaded on 1-D gels for a Western blot using an anti-p-IGF-1R β antibody. The membranes were then stripped and re-probed with a total IGF-1Rβ antibody. The data were represented normalized to the control with Mean + SEM, analyzed with a one-way ANOVA, and corrected with a Bonferroni multiple comparison test. Means without a common letter differ are considered significant, *p* < 0.05.

**Figure 7 ijms-24-07273-f007:**
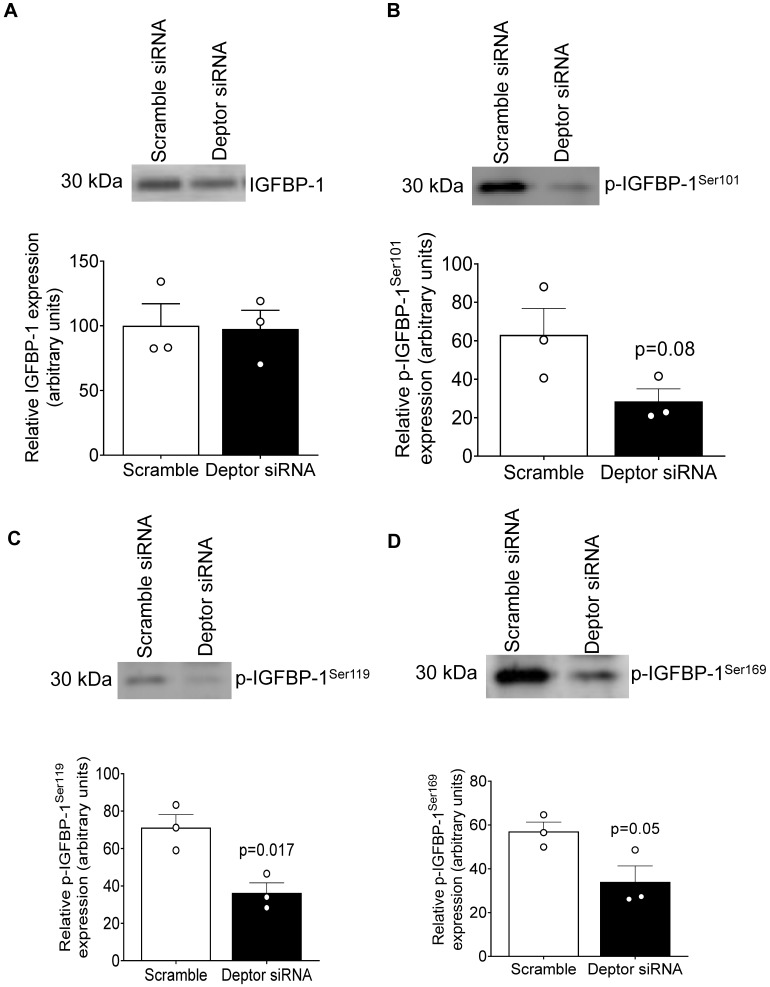
IGFBP-1 secretion and phosphorylation in CM from HepG2 cells treated with CM from PHT cells with the control and *DEPTOR* siRNA. Equal aliquots of CM from HepG2 cells (7 μL) treated with CM from PHT cells with the control or *DEPTOR* siRNA (activates mTOR signaling) were loaded on 1-D gels for a Western blot using anti-human IGFBP-1 monoclonal antibody (mAb 6303) (**A**); 40 μL of cell media loaded for pS101 (**B**), pS119 (**C**), and pS169 (**D**). Samples were normalized to non-targeting siRNA control, and data were represented as Mean + SEM in bar figures and analyzed with unpaired t-test; *p* < 0.05 was considered significant.

**Figure 8 ijms-24-07273-f008:**
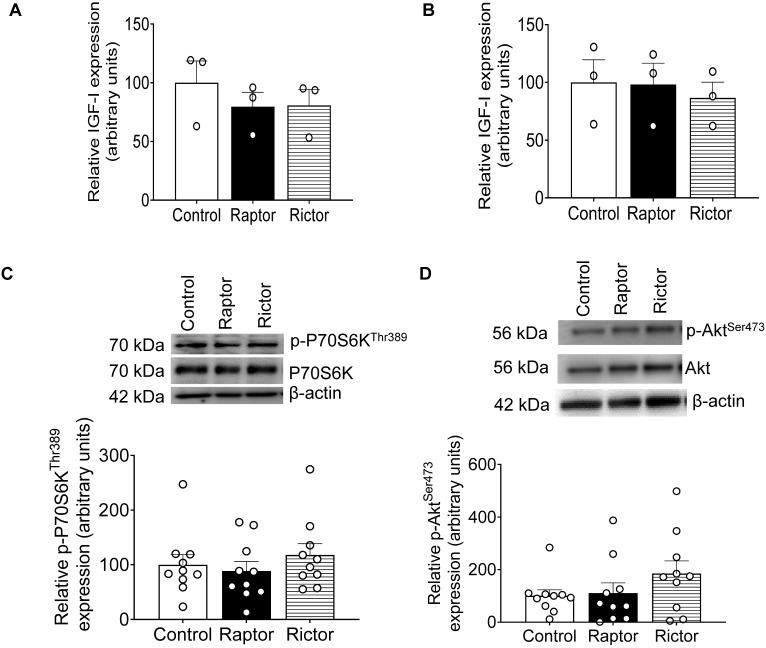
ELISA analysis of IGF-1 secretion before and after treatment of HepG2 cells with CM from PHT cells with the control, *RAPTOR*, and *RICTOR* siRNA and densitometric analysis of mTORC1 and mTORC2 functional readouts from cell lysates of HepG2 treated with CM from PHT cells with the control, *RAPTOR*, and *RICTOR* siRNA. IGF-1 secretion was determined in CM from PHT cells with the control, *RAPTOR*, and *RICTOR* siRNA before treatment of HepG2 cells (**A**) and CM from HepG2 cells after treatment with CM from PHT cells with the control, *RAPTOR,* and *RICTOR* (**B**). Equal aliquots (50 μL) of CM were analyzed with an IGF-1 ELISA kit. A representative Western blot of HepG2 cell lysates (35 μg protein) displaying p-P70S6K ^Thr389^ (**C**) and p-Akt ^Ser473^ (**D**) in HepG2 cells treated with CM from PHT cells with the control, *RAPTOR,* or *RICTOR* siRNA. Samples were normalized to the control. The data were represented as Mean + SEM in bar figures and analyzed with a one-way ANOVA and Bonferroni Multiple Comparison tests; *p* < 0.05 was considered significant.

**Figure 9 ijms-24-07273-f009:**
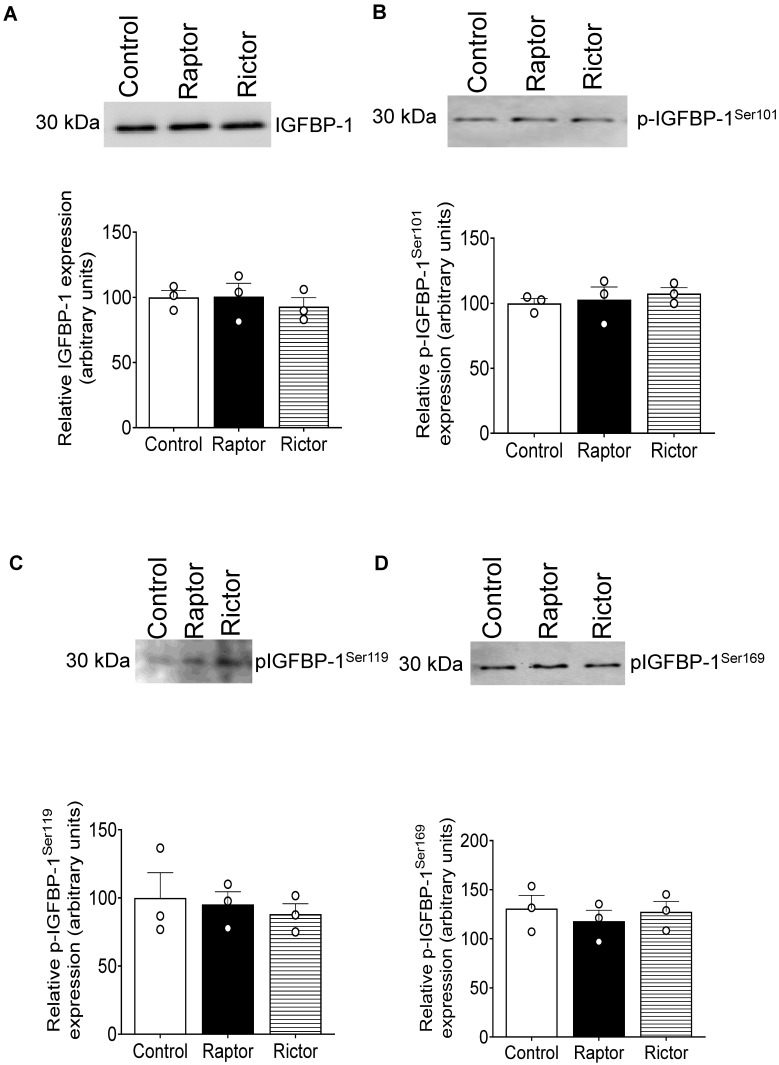
Analysis of IGFBP-1 secretion and phosphorylation from HepG2 cells after treatment by Renal Carcinoma cells transfected with the control, *RAPTOR*, and *RICTOR* siRNA. Equal aliquots of HepG2 cell media (7 μL) were loaded on 1-D gels for a Western blot using anti-human IGFBP-1 monoclonal antibody (mAb 6303) (**A**); 40 μL loaded for pSer101 (**B**), pSer119 (**C**), and pSer169 (**D**). Samples were normalized to the control. The data were represented as Mean + SEM in bar figures and analyzed with a one-way ANOVA and Bonferroni Multiple Comparison tests; *p* < 0.05 was considered significant.

**Figure 10 ijms-24-07273-f010:**
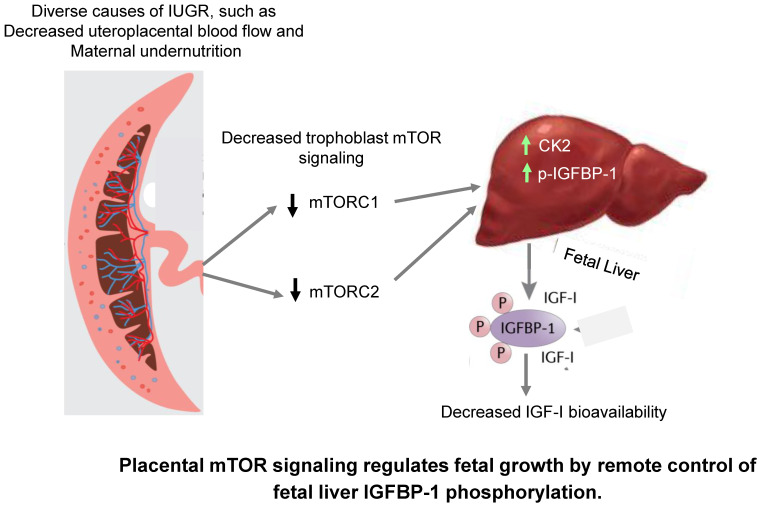
The proposed model of placental mTOR signaling directly influences the endocrine and metabolic function of fetal tissues. Specifically, the bioavailability of IGF-1 is mediated by signals released into fetal circulation.

## Data Availability

All the data available on request.

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
