# Peer review of "Placental Remote Control of Fetal Metabolism: Trophoblast mTOR Signaling Regulates Liver IGFBP-1 Phosphorylation and IGF-1 Bioavailability"

_ijms, 2023, doi:10.3390/ijms24087273_

Round 1
Reviewer 1 Report
It is a good job. The figures are easy to understand and the tecniques used are good explained. Maybe I would like to read a long discussion were the authors explain the meaning of the results in a complete organism and not write again the most of the time results and tecniques. What would your results be useful in clinical? What are the limitations of your study?
Why did you only used control mothers?
Author Response
Reviewer 1:
It is a good job. The figures are easy to understand, and the techniques used are good explained. Maybe I would like to read a long discussion where the authors explain the meaning of the results in a complete organism and not write most of the time results and techniques again.
Response: We thank the reviewer for the positive comments. In the discussion, we have added some additional text to better explain what the results mean and removed some wording related to repeating the results and discussing the methods used.
What would your results be useful in clinical? What are the limitations of your study?
Response: We provide evidence that placental mTOR signaling may control fetal growth by modulating the phosphorylation of IGFBP-1 in the fetal liver. Because the placenta - in contrast to the fetus – is easily accessible from the maternal circulation and an array of approaches to specifically target the placenta, for example, nanoparticles currently under development, we speculate that gene targeting of placental mTOR may provide an avenue to correct abnormal fetal growth in the future. The main limitation of the study is that the data is based on in vitro studies, and our findings need to be confirmed in vivo using the placental-specific targeting of mTOR. We have added these comments to the Discussion.
Why did you only use control mothers?
Response: We thank the reviewer for this insightful comment. Indeed, we could have used the conditioned media collected from PHT cells isolated from pregnancies complicated by IUGR. However, we believe that the design we elected to use (mTOR gene targeting in PHT cells from control mothers) provides very strong proof-of-principle studies for the novel mechanism involving mTOR ‘remotely’ controlling fetal liver IGFBP-1 phosphorylation. In addition, adding the IUGR PHT cell arm to this study is a major undertaking that we have elected to save for future studies.
Reviewer 2 Report
This work investigates the role of trophoblast mTOR signaling on the availability of the fetal growth factor IGF-1 through its specific receptor signaling in the context of IUGR, a relevant pregnancy complication. The effect of placental function on fetal development is a trending topic in placental research, and the field is expected to grow substantially in the years to come. This work, despite the obvious limitation of the chosen in vitro model, contributes to the understanding of the remote modulation of biological processes by placental factors. Overall, the quality of writing, experimental design, and execution is excellent, leaving only minor issues to be addressed.
Introduction:
Line 70: Reference 15 refers only to placental malaria, using the term infections is too broad. Please add references to support the use of infections in general or change the text to include only malaria infection.
Results.
Lines 274-276. The grammar in this paragraph could be clearer. I think the words mTORC1 and RAPTOR are mixed up, please check this issue.
Line 303: The selected subtitle does not follow the same pattern as the other titles, please use a descriptive subtitle.
Line 316: Please include the p-value for the RIC condition in the figure.
Figures
Figure 5A. Please use a larger font for X and Y axis, the size is very hard to read.
Figure 5B. P value for Ric is missing
Author Response
Reviewer 2:
This work investigates the role of trophoblast mTOR signaling on the availability of the fetal growth factor IGF-1 through its specific receptor signaling in the context of IUGR, a relevant pregnancy complication. The effect of placental function on fetal development is a trending topic in placental research, and the field is expected to grow substantially in the years to come. This work, despite the obvious limitation of the chosen in vitro model, contributes to the understanding of the remote modulation of biological processes by placental factors. Overall, the quality of writing, experimental design, and execution is excellent, leaving only minor issues to be addressed.
Response: We thank the reviewer for the positive comments.
Line 70: Reference 15 refers only to placental malaria, using the term infections is too broad. Please add references to support the use of infections in general or change the text to include only malaria infection.
Response: As suggested by the reviewer, we have now incorporated the term “malaria infection” in Line # 90.
Results. Lines 274-276. The grammar in this paragraph could be clearer. I think the words mTORC1 and RAPTOR are mixed up, please check this issue.
Response: We apologize for the imperfect wording, and we have now restructured the sentence as suggested by the reviewer.
Line 303: The selected subtitle does not follow the same pattern as the other titles; please use a descriptive subtitle.
Response: We have now modified the subtitle as suggested by the reviewer.
Line 316: Please include the p-value for the RIC condition in the figure.
Response: we have now incorporated the p-value for the RICTOR condition (Figure 5 B).
Figure 5A. Please use a larger font for X and Y axis, the size is very hard to read.
Response: We have now increased the X and Y axis font size according to the reviewer’s suggestion
Figure 5B. P value for Ric is missing.
Response: We have now incorporated the p-value for the RICTOR condition (Figure 5 B).
Reviewer 3 Report
In this paper authors described a potential mechanism related to intrauterine growth restriction (IUGR). There results showed that decreasing mTOR signal in trophoblast can increase IGFBP-1 phosphorylation by using conditional media from trophoblast cells treated or not with mTOR SiRNA .
Some questions need to be addressed for better understanding of the results:
Fig 2A: Does CM from different condition affect viability/proliferation of HepG2 cells. Quantification of IGFBP-1 should be reported to total amount of protein of HepG2 cell lysate.
Fig 3A-C: based on in Fig 2A increased phosphorylation of IGFBP might be due to increase in total IGFBP expression and not phosphorylation itself, hence it is good to normalize phosphor expression to total amount of IGFBP to better conclude.
Fig5: In order to validate implication of CK2 in IGFBP-1 phosphorylation, additional experiment using SiRNA and or CK2 inhibitor is needed to show a decrease in IGFBP-1 phosphorylation under those conditions.
Fig7: In contrast to author’s statement western blot image showed a decrease in IGFBP-1 after Deptor silencing, quantification should also be normalized to total amount of protein.
Author Response
Fig 2A: Does CM from different condition affect viability/proliferation of HepG2 cells?
Response: To ensure cell viability from CM from different treatments, we employed a trypan blue exclusion assay. Cells were counted using the Countess Automated Cell Counter (Life Technologies, Carlsbad, CA). Cell survival was determined as a ratio of live/total cells from different conditions, which remained comparable. The data is not presented in the paper.
Quantification of IGFBP-1 should be reported to total amount of protein of HepG2 cell lysate.
Response: The strategy proposed by the reviewer may be applicable to most phosphoproteins synthesized by the cells. Because IGFBP-1 is a secretory protein, the amount of IGFBP-1 quantified to total cell lysate protein will not be biologically justifiable.
In the absence of a recognized loading control for secreted proteins (equivalent to actin for non-secreted proteins), we used an equal volume of cell media 1,2 as a loading control. We have now incorporated this brief description in the revised manuscript methods section.
Fig 3A-C: based on in Fig 2A increased phosphorylation of IGFBP might be due to increase in total IGFBP expression and not phosphorylation itself, hence it is good to normalize phosphor expression to total amount of IGFBP to better conclude.
Response: We strongly argue that correcting IGFBP-1 phosphorylation for total IGFBP-1 will be misleading. The reason being that the IGFBP-1 phospho/total ratio often provides inaccurate information with respect to the biological effects. Because the phosphorylation of IGFBP-1 dramatically increases the affinity for IGF-I (see further below), changes in phosphorylation will have a more powerful impact on the biological effect than changes in total IGFBP-1 expression.
For instance, the decrease in phospho/total ratio to half would be interpreted as a marked decrease in biological effect. This line of reasoning is strongly supported by experimental data. Our previous ELISA data 1 revealed that the total serine phosphorylation of IGFBP-1 increased for hypoxia and leucine deprivation (~2 to 2.5-fold), proportional to the overall increases in IGFBP-1 secretion (total IGFBP-1 ELISA ~2 to 2.5-fold) (Seferovic MD et al., 20091, Figure 2A and B). Thus, the phospho/total ratio is unchanged. Using LC-MS analysis, in hypoxia, pS169 showed ~ 4-fold increase in p-peptide peak signal intensity relative to the control. In leucine deprivation, p-peptide intensity for pS119 was increased to similar levels ~ 4-fold (Seferovic MD et al., 20091, Table 1). Importantly, despite the unchanged phospho/total ratio, BiaCore analysis indicated the hyperphosphorylated IGFBP-1 isoforms found in hypoxia caused ~300-fold greater affinity for IGF-1 relative to the IGFBP-1 from the controls while leucine deprivation treated cells showed ~30-fold increase in IGF-I affinity (Seferovic MD et al., 20091, Table 2). The effectiveness of phosphorylation at specific residues on IGFBP-1 for their ability to differentially alter the capacity of IGF-1 to bind and stimulate IGF-IR signaling was further confirmed in another study using site directed mutagenesis 3.
Collectively, these findings suggest that the predominant increase in IGF-I affinity is due to an increase in the abundance of specific phospho-IGFBP-1 isoform (site and degree of phosphorylation) rather than increases in total IGFBP-1 or even total phospho-IGFBP-1. We therefore expect that the predominant changes in IGF-I bioavailability are not based on total IGFBP-1 or levels of total phospho-IGFBP-1 but due to combined effects of sites and degree of phosphorylation.
Fig5: In order to validate implication of CK2 in IGFBP-1 phosphorylation, additional experiment using SiRNA and or CK2 inhibitor is needed to show a decrease in IGFBP-1 phosphorylation under those conditions.
Response: The reviewer raises an important point suggesting additional siRNA and/or CK2 inhibitor studies. We would like to draw reviewer’s attention to our prior studies1,4, where using cultured HepG2 cells we applied both these approaches for validation of CK2 as a kinase involved in IGFBP-1 phosphorylation.
In brief, to confirm that CK2 plays an important role in mediating IGFBP-1 phosphorylation, we performed silencing of the CK2 holoenzyme by targeting the two catalytic subunits (CK2α or CK2α′) as well as the regulatory subunit (CK2β) in combination. which resulted in a marked decrease in IGFBP-1 phosphorylation at all three phosphorylation sites, Ser101, Ser119, and Ser1694,5. In contrast to the effect on IGFBP-1 phosphorylation, CK2 holoenzyme silencing did not alter total protein expression of IGFBP-1 secreted by HepG2 cells. We further determined the effects of pharmacological CK2 inhibitor 4,5,6,7-tetrabromobenzotriazole (TBB) which decreased IGFBP-1 phosphorylation in dose dependent manner.
Another supporting evidence is the consensus sequence for CK2 phosphorylation which requires an acidic residue in position downstream from the phosphoacceptor site and is generally accompanied by additional acidic residues. Two of the three IGFBP-1 phosphorylation sites (Ser119 and Ser169) in IGFBP-1 conform precisely to this general recognition motif. We also validated the role of CK2 using three synthetic non-phosphorylated IGFBP-1 peptides and MRM-MS strategy. We demonstrate that CK2 can directly phosphorylate IGFBP-1 at all three serine residues. These results are consistent between two independent approaches: immunoblotting and MRM-MS analyses4.
The biological effects of pharmacological inhibition of CK2 which reduced IGFBP-1 phosphorylation led to also reduced IGF-1 receptor (IGF-1Rb) autophosphorylation4.
Together data from these studies and several other studies from our laboratory 1-5 allowed us to gain clear evidence for a CK2-mediated mechanism for regulation of IGFBP-1 hyperphosphorylation.
Fig 7: In contrast to author’s statement western blot image showed a decrease in IGFBP-1 after Deptor silencing, quantification should also be normalized to total amount of protein.
Response: We thank the reviewer and apologize for this error. We have made changes in the manuscript text as follows “Phosphorylation was markedly inhibited at all three (Ser101, 119, and 169) phosphorylation sites in HepG2 cells treated with CM from PHT cells with DEPTOR silencing while relatively smaller decrease in IGFBP-1 secretion (Figure 7A-D)”.
In regards to normalization/quantitation, as we mentioned earlier, in the absence of a recognized loading control for secreted proteins (equivalent to actin for non-secreted proteins), we used an equal volume of cell media 1,2 as a loading control. We have now incorporated this brief description in the revised manuscript methods section.
1 Seferovic, M. D. et al. Hypoxia and leucine deprivation induce human insulin-like growth factor binding protein-1 hyperphosphorylation and increase its biological activity. Endocrinology 150, 220-231, doi:10.1210/en.2008-0657 (2009).
2 Damerill, I. et al. Hypoxia Increases IGFBP-1 Phosphorylation Mediated by mTOR Inhibition. Mol Endocrinol 30, 201-216, doi:10.1210/me.2015-1194 (2016).
3 Abu Shehab, M., Iosef, C., Wildgruber, R., Sardana, G. & Gupta, M. B. Phosphorylation of IGFBP-1 at discrete sites elicits variable effects on IGF-I receptor autophosphorylation. Endocrinology 154, 1130-1143, doi:10.1210/en.2012-1962 (2013).
4 Malkani, N. et al. Increased IGFBP-1 phosphorylation in response to leucine deprivation is mediated by CK2 and PKC. Mol Cell Endocrinol 425, 48-60, doi:10.1016/j.mce.2015.12.006 (2016).
5 Abu Shehab, M. et al. Liver mTOR controls IGF-I bioavailability by regulation of protein kinase CK2 and IGFBP-1 phosphorylation in fetal growth restriction. Endocrinology 155, 1327-1339, doi:10.1210/en.2013-1759 (2014).
Reviewer 4 Report
This study, by Dr. Rosario et al, is centered around the role of the placenta in regulating fetal metabolism and growth and demonstrates that mTOR signaling in the trophoblast directly regulates fetal growth by modulating the bioavailability of IGF-1. Since there are few descriptions about Result overall, I felt that it would be easier for the reader to understand if the authors describ Result Section a little more carefully. The notation in the figure is not uniform (eg. Rap or Raptor), so please check it. Also, I have some comments, which should be addressed.
The authors used PHT collected from healthy term with silencing of mTOR complexes. Does mTOR inhibition occur in placenta of FGR? Moreover, have the authors check CM derived from placenta of FGR decrease IGF-1 bioavailability?
In Figure 4, what does pSer169/pSer174 level change mean? An explanation should be added. Why is pSer169/pSer174 level increased in the Ric-treated group, unlike Figure 3?
In Figure 5, why is CK2 level increased only in the Ric-treated group? Moreover, although CK2 is a phosphorylation enzyme of IGFBP-1, CK2 activity has a smaller fluctuation range than the phosphorylation rate of IGFBP-1. How should this be interpreted?
Author Response
Reviewer 3:
This study, by Dr. Rosario et al, is centered around the role of the placenta in regulating fetal metabolism and growth and demonstrates that mTOR signaling in the trophoblast directly regulates fetal growth by modulating the bioavailability of IGF-1.
Since there are few descriptions about Result overall, I felt that it would be easier for the reader to understand if the authors described Result Section a little more carefully. The notation in the figure is not uniform (eg. Rap or Raptor), so please check it. Also, I have some comments, which should be addressed.
Response: We have now made the notation in the figure’s uniform and added some additional detail to the description of the results
The authors used PHT collected from healthy term with silencing of mTOR complexes. Does mTOR inhibition occur in placenta of FGR? Moreover, have the authors check CM derived from placenta of FGR decrease IGF-1 bioavailability?
Response: We and others demonstrated that placental mTOR signaling is inhibited in IUGR in women 1-5 and animal models of IUGR 6-9. Moreover, we previously demonstrated that IGFBP-1 is hyperphosphorylated in both maternal and fetal plasma in human IUGR pregnancies 10, which inhibits IGF1 secretion in FGR11. We believe that the design we elected to use (mTOR gene targeting in PHT cells from control mothers) provides very strong proof-of-principle studies for the novel mechanism involving mTOR ‘remotely’ controlling fetal liver IGFBP-1 phosphorylation. In addition, adding an IUGR PHT cell arm to this study is a major undertaking that we have elected to save for future studies.
In Figure 4, what does pSer169/pSer174 level change mean? An explanation should be added. Why is pSer169/pSer174 level increased in the Ric-treated group, unlike Figure 3?
Response: We have previously shown Ser174 phosphorylation singly and/or combined with Ser169 is an mTOR-sensitive IGFBP-1 phosphorylation potentially involved in IGF-1 binding 12 In the current study, increased phosphorylation of IGFBP-1 with RICTOR silencing confirms the involvement of Ser174 phosphorylation combined with Ser169 and suggests that phosphorylation at residue Ser174 likely affects the affinity of the IGFBP-1 to bind IGF-112.
We believe that the differences between data in Fig 3 and 4 are due to the use of different antibodies. Thus, in Figure 3, an antibody specifically directly to IGFBP-1 when phosphorylated at Ser169 was used, and this antibody is not expected to recognize when both Ser169/Ser174 are phosphorylated. Conversely, In Fig 4, an antibody targeting IGFBP-1 only when dually phosphorylated at Ser169/Ser174 was used12.
In Figure 5, why is CK2 level increased only in the Ric-treated group? Moreover, although CK2 is a phosphorylation enzyme of IGFBP-1, CK2 activity has a smaller fluctuation range than the phosphorylation rate of IGFBP-1. How should this be interpreted?
Response: Although we have provided evidence previously that CK2 is a major contributor to IGFBP-1 phosphorylation 10,13-17, including in response to changes in mTOR signaling 10, we often find that measured changes in CK activity are lower than the measured changes in IGFBP-1 phosphorylation 15. There may be several reasons for this discrepancy. First, it is possible that the relationship between CK activity and IGFBP-1 phosphorylation is not 1:1, and that small changes in CK2 activity are related to larger changes in IGFBP-1 phosphorylation. Second, there may be additional kinases, such as PKA or PKC 13,15,17, contributing to IGFBP-1 phosphorylation; third, it cannot be excluded that our approach to measure CK2 activity underestimates the actual activity
References
1 Roos, S. et al. Mammalian target of rapamycin in the human placenta regulates leucine transport and is down-regulated in restricted fetal growth
J Physiol 582, 449-459 (2007).
2 Yung, H. W. et al. Evidence of translation inhibition and endoplasmic reticulum stress in the etiology of human intrauterine growth restriction. Am J Pathol 173, 311-314 (2008).
3 Chen, Y. Y. et al. Increased ubiquitination and reduced plasma membrane trafficking of placental amino acid transporter SNAT-2 in human IUGR. Clin Sci (Lond) 129, 1131-1141, doi:10.1042/cs20150511 (2015).
4 Hung, T. H. et al. Mammalian target of rapamycin signaling is a mechanistic link between increased endoplasmic reticulum stress and autophagy in the placentas of pregnancies complicated by growth restriction. Placenta 60, 9-20, doi:10.1016/j.placenta.2017.10.001 (2017).
5 Dimasuay, K. G. et al. Inhibition of placental mTOR signaling provides a link between placental malaria and reduced birthweight. BMC Med 15, 1, doi:10.1186/s12916-016-0759-3 (2017).
6 Rosario, F. J. et al. Maternal protein restriction in the rat inhibits placental insulin, mTOR, and STAT3 signaling and down-regulates placental amino acid transporters. Endocrinology 152, 1119-1129, doi:10.1210/en.2010-1153 (2011).
7 Kavitha, J. V. et al. Down-regulation of placental mTOR, insulin/IGF-I signaling, and nutrient transporters in response to maternal nutrient restriction in the baboon. FASEB J 28, 1294-1305, doi:10.1096/fj.13-242271 (2014).
8 Rosario, F. J. et al. Chronic maternal infusion of full-length adiponectin in pregnant mice down-regulates placental amino acid transporter activity and expression and decreases fetal growth. J Physiol 590, 1495-1509 (2012).
9 Rosario, F. J., Nathanielsz, P. W., Powell, T. L. & Jansson, T. Maternal folate deficiency causes inhibition of mTOR signaling, down-regulation of placental amino acid transporters and fetal growth restriction in mice. Sci Rep 7, 3982, doi:10.1038/s41598-017-03888-2 (2017).
10 Abu Shehab, M. et al. Liver mTOR controls IGF-I bioavailability by regulation of protein kinase CK2 and IGFBP-1 phosphorylation in fetal growth restriction. Endocrinology 155, 1327-1339, doi:10.1210/en.2013-1759 (2014).
11 Lassarre, C. et al. Serum insulin-like growth factors and insulin-like growth factor binding proteins in the human fetus. Relationships with growth in normal subjects and in subjects with intrauterine growth retardation. Pediatr Res 29, 219-225, doi:10.1203/00006450-199103000-00001 (1991).
12 Damerill, I. et al. Hypoxia Increases IGFBP-1 Phosphorylation Mediated by mTOR Inhibition. Mol Endocrinol 30, 201-216, doi:10.1210/me.2015-1194 (2016).
13 Abu Shehab, M. et al. Inhibition of decidual IGF-1 signaling in response to hypoxia and leucine deprivation is mediated by mTOR and AAR pathways and increased IGFBP-1 phosphorylation. Mol Cell Endocrinol 512, 110865, doi:10.1016/j.mce.2020.110865 (2020).
14 Kakadia, J. H. et al. Hyperphosphorylation of fetal liver IGFBP-1 precedes slowing of fetal growth in nutrient-restricted baboons and may be a mechanism underlying IUGR. Am J Physiol Endocrinol Metab 319, E614-E628, doi:10.1152/ajpendo.00220.2020 (2020).
15 Malkani, N. et al. Increased IGFBP-1 phosphorylation in response to leucine deprivation is mediated by CK2 and PKC. Mol Cell Endocrinol 425, 48-60, doi:10.1016/j.mce.2015.12.006 (2016).
16 Nandi, P. et al. Mechanistic Target of Rapamycin Complex 1 Signaling Links Hypoxia to Increased IGFBP-1 Phosphorylation in Primary Human Decidualized Endometrial Stromal Cells. Biomolecules 11, doi:10.3390/biom11091382 (2021).
17 Chen, A. W. et al. IGFBP-1 hyperphosphorylation in response to nutrient deprivation is mediated by activation of protein kinase Calpha (PKCalpha). Mol Cell Endocrinol 536, 111400, doi:10.1016/j.mce.2021.111400 (2021).
Round 2
Reviewer 4 Report
>Response: We have previously shown Ser174 phosphorylation singly and/or combined with Ser169 is an mTOR-sensitive IGFBP-1 phosphorylation potentially involved in IGF-1 binding. In the current study, increased phosphorylation of IGFBP-1 with RICTOR silencing confirms the involvement of Ser174 phosphorylation combined with Ser169 and suggests that phosphorylation at residue Ser174 likely affects the affinity of the IGFBP-1 to bind IGF-1.
> We believe that the differences between data in Fig 3 and 4 are due to the use of different antibodies. Thus, in Figure 3, an antibody specifically directly to IGFBP-1 when phosphorylated at Ser169 was used, and this antibody is not expected to recognize when both Ser169/Ser174 are phosphorylated. Conversely, In Fig 4, an antibody targeting IGFBP-1 only when dually phosphorylated at Ser169/Ser174 was used.
Western blot for pSer174 should be shown. For Ser174 phosphorylation, an explanation should be added as to what the presence or absence of a combine of Ser169 means.
The following should be added in Discussion Section.
> Response: Although we have provided evidence previously that CK2 is a major contributor to IGFBP-1 phosphorylation, including in response to changes in mTOR signaling, we often find that measured changes in CK activity are lower than the measured changes in IGFBP-1 phosphorylation. There may be several reasons for this discrepancy. First, it is possible that the relationship between CK activity and IGFBP-1 phosphorylation is not 1:1, and that small changes in CK2 activity are related to larger changes in IGFBP-1 phosphorylation. Second, there may be additional kinases, such as PKA or PKC, contributing to IGFBP-1 phosphorylation; third, it cannot be excluded that our approach to measure CK2 activity underestimates the actual activity.
Author Response
Western blot for pSer174 should be shown.
Response: No antibodies are available to assess this modification, therefore, western could not be done for pSer174. Considering this limitation with Western blot analysis, targeted MRM-MS normalized signals with a standard peptide within IGFBP-1 (internal peptide) yielded highly precise, reliable, and reproducible data.
For Ser174 phosphorylation, an explanation should be added as to what the presence or absence of a combine of Ser169 means.
Response
As suggested by the reviewers we have incorporated the following sentences in the discussion part of the manuscript.
"Although we have provided evidence previously that CK2 is a major contributor to IGFBP-1 phosphorylation, including in response to changes in mTOR signaling, we often find that measured changes in CK2 activity are lower than the measured changes in IGFBP-1 phosphorylation. There may be several reasons for this discrepancy. First, it is possible that the relationship between CK2 activity and IGFBP-1 phosphorylation is not 1:1, and that small changes in CK2 activity are related to larger changes in IGFBP-1 phosphorylation. Second, there may be additional kinases, such as PKA or PKC, contributing to IGFBP-1 phosphorylation; third, it cannot be excluded that our approach to measure CK2 activity underestimates the actual activity"
Response: In order to explain the significance, we refer the reviewer to our previous study, where MRM-MS analysis allowed us to distinguish between the sole pSer169 and the combined tandem phosphorylation of Ser169 together with Ser174. Due to the lack of pSer174-specific antibody, we were unable to determine whether the total phosphorylation signal on Ser169 detected by Western blotting is cumulative of the 2 phosphorylation events on Ser169/Ser174 peptide.
To explore the structure-functional significance of dual-site phosphorylation, we mapped the proximity for pSer174 binding with IGF-I using molecular modeling where IGFBP-1 protein was modeled (in silico) in complex with IGF-I based on the known crystal structures. Structural modeling illustrated the relative proximity of Ser174 and Ser169 with IGF-I in the thyroglobulin type-I (TY) domain of IGFBP-1. Ser174 in the IGFBP-1-IGF-I complex model was found to be near regions of IGFBP-1 involved in IGF-I binding, providing support for a close interaction of Ser174 with the structured regions of IGFBP-1 involved in IGF-I binding. Although structure-functional proof is currently lacking, we speculate that increased phosphorylation at Ser169 together with Ser174 at the C-terminal end may cooperatively/synergistically function to increase the binding affinity of IGFBP-1 to IGF-I through much stronger interactions. This effect, in tandem with other IGFBP-1 phosphorylation sites in the linker region, could potentially prolong the binding of IGF-I to IGFBP-1 and enhance the function of IGFBP-1 in reducing IGF-I bioavailability1,2.
- Damerill et al., Hypoxia Increases IGFBP-1 Phosphorylation Mediated by mTOR Inhibition. Mol Endocrinol 30, 201-216 593
(2016).
- M. B. Gupta et al., IUGR Is Associated With Marked Hyperphosphorylation of Decidual and Maternal Plasma IGFBP-1. J 566 J. Clin Endocrinol Metab 104, 408-422 (2019).